# LLM4Rerank: LLM-based Auto-Reranking Framework for Recommendations

Submission Id: 805

## Abstract

Reranking is significant for recommender systems due to its pivotal role in refining recommendation results. Numerous reranking models have emerged to meet diverse reranking requirements in practical applications, which not only prioritize accuracy but also consider additional aspects such as diversity and fairness. However, most of the existing models struggle to strike a harmonious balance between these diverse aspects at the model level. Additionally, the scalability and personalization of these models are often limited by their complexity and a lack of attention to the varying importance of different aspects in diverse reranking scenarios. To address these issues, we propose LLM4Rerank, a comprehensive LLM-based reranking framework designed to bridge the gap between various reranking aspects while ensuring scalability and personalized performance. Specifically, we abstract different aspects into distinct nodes and construct a fully connected graph for LLM to automatically consider aspects like accuracy, diversity, fairness, and more, all in a coherent Chain-of-Thought (CoT) process. To further enhance personalization during reranking, we facilitate a customizable input mechanism that allows fine-tuning of LLM's focus on different aspects according to specific reranking needs. Experimental results on three widely used public datasets demonstrate that LLM4Rerank outperforms existing state-of-the-art reranking models across multiple aspects. The implementation code is available for reproducibility [1].

**Relevance Statement**: This paper presents an LLM-based auto-reranking framework that integrates various personalized aspects during the reranking stage of a recommender system. This is highly pertinent to the Web track topic **user modeling, personalization, and recommendation**. Moreover, the proposed method can be directly used to conduct reranking with personalized requirements, which is closely related to the applications of the Web.

## CCS Concepts

• **Information systems** → **Retrieval models and ranking**; Data mining.

---

[1]https://anonymous.4open.science/r/LLM4Rerank-5EA2

---

## Keywords

Reranking, Recommender System, Large Language Model

**ACM Reference Format:**
Anonymous Author(s). 2025. LLM4Rerank: LLM-based Auto-Reranking Framework for Recommendations. In *Proceedings of Make sure to enter the correct conference title from your rights confirmation emai (WWW '25)*. ACM, New York, NY, USA, 11 pages. https://doi.org/XXXXXXX.XXXXXXX

## 1 Introduction

Reranking is a fundamental technique in the field of recommender systems [21]. Its importance lies in its ability to refine and enhance the results generated by ranking models, ultimately providing users with the most relevant and personalized recommendations. In the typical recommendation process, interaction features (e.g., user attributes and item properties) are initially utilized by a ranking model to generate a candidate list for the user. Subsequently, a reranking model is applied to further scrutinize the association between candidate items, delving into the nuances and intricacies of user preferences, and ultimately producing the final recommendation list. Currently, existing reranking models predominantly focus on improving the accuracy aspect of recommendations [1, 19, 24]. Although accuracy is crucial, it is equally important to consider extensive aspects of the recommended list in practical use, such as diversity [4, 6, 39] and fairness [36, 41]. Diversity ensures that users are exposed to a varied range of items, while fairness guarantees equal representation and exposure of different categories or sellers. Though several research [4, 6, 36] try to combine one of them with the accuracy aspect for modeling, how to better consider and balance more aspects simultaneously remains a problem.

Specifically, existing reranking models do suffer from several limitations. Firstly, it is challenging to comprehensively consider and balance the complex combination of multiple aspects in modeling, primarily due to the substantial semantic gap between these aspects [8, 32]. This is because each aspect scrutinizes the recommendation list via unique attribute dimensions, highlighting intricate semantic relationships and distinctions. This complexity underscores the substantial gap that exists between different aspects. Furthermore, scalability issues present another major hurdle, inhibiting the application of a singular model across diverse recommendation settings that may prioritize different aspects or functional rules. This challenge is particularly pronounced when introducing novel aspects or custom reranking rules, such as backward rules or stop conditions, not initially anticipated during the model's development. Moreover, the inability to personalize the amalgamation of various aspects further limits the personalization of existing models, as noted in prior research [25]. Once deployed, the output tendency of a specific model on different aspects is fixed and cannot be intelligently adjusted according to evolving business or user preferences.

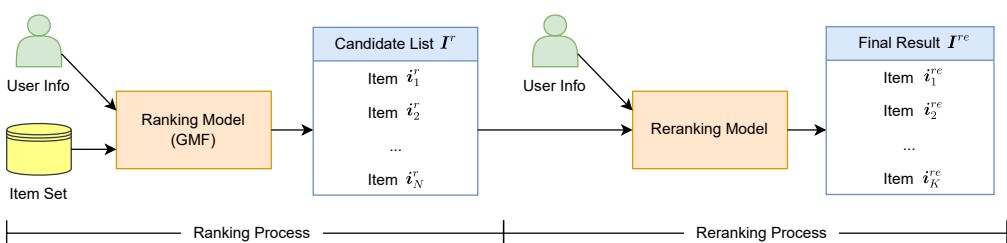

**Figure 1: Ranking and reranking process in recommendations.**

To overcome these barriers, an optimal solution would involve creating a versatile reranking framework capable of simultaneously accounting for diverse aspect combinations and semantic nuances. Such a framework would flexibly accommodate the unique demands of different contexts and user needs, offering a more dynamic and tailored reranking solution.

Currently, with the rapid development of Large Language Models (LLMs) [9, 23], extensive research has been undertaken to evaluate the capabilities of LLMs in various contexts [5, 34]. Previous studies [17, 22, 28] highlight that zero-shot LLMs may not achieve the same level of proficiency as specialized models in tasks such as information extraction and recommendation. This limitation is often attributed to their restricted token count and difficulties in processing extensive contexts that include thousands of items [20]. Despite these challenges, these studies have shown that LLMs can perform comparably or even surpass supervised benchmarks in reranking tasks. This efficacy is credited to their robust semantic understanding capabilities within concise contexts involving a limited number of items. Thus, the application of LLMs in the reranking phase of recommendations could serve as a key strategy to merge different aspects by enhancing semantic understanding.

Nonetheless, several significant challenges arise when modeling a reranking framework using LLMs. The first challenge involves ensuring the scalability of the framework in a well-organized and flexible manner, allowing it to accommodate current aspect requirements while also remaining adaptable to potential future aspects. The second challenge revolves around formulating a mechanism capable of automatically combining diverse aspect requirements in accordance with specific recommendation settings or user preferences, ultimately achieving genuine personalization.

To address these challenges, we propose LLM4Rerank, an innovative reranking framework that harnesses the power of zero-shot LLMs for more precise reranking. Specifically, LLM4Rerank represents various aspect requirements in reranking as distinct nodes, allowing the framework to automatically incorporate these nodes in a Chain-of-Thought (CoT) manner [3, 35, 40]. The advantage of this approach is twofold: it ensures scalability, allowing for the seamless inclusion of new nodes to address emerging aspect requirements. To demonstrate this, in addition to the accuracy aspect, diversity, and fairness aspects are also added for LLM4Rerank modeling in this paper. Additionally, the LLM used in LLM4Rerank can automatically determine the next node to consider, guided by the current reranking history and an additional sentence input referred to as the "Goal", which is provided by the user or deployer, representing the overall focus and objective of the ongoing reranking process. This dynamic process enables LLM4Rerank to achieve enhanced personalization in the reranking process.

In summary, in this paper, our contributions could be summarized as follows:

- To the best of our knowledge, this work is the first endeavor to automatically integrate multiple aspects and could thus measure different aspects in a unified semantic space comprehensively through a multi-hop reranking procedure employing LLMs.
- We propose LLM4Rerank, a novel framework that can handle the complex combination of various aspect requirements, such as accuracy, diversity, and fairness, within the reranking process. LLM4Rerank offers the potential for superior performance, scalability, and personalization in reranking.
- Experiments conducted on three widely used industrial datasets demonstrate that LLM4Rerank outperforms existing baselines in all aspects considered. This validates its efficacy and superiority in enhancing performance, scalability and personalization within the reranking process of recommender systems.

## 2 Framework

This section outlines the problem formulation for the reranking task in recommendations, followed by a comprehensive overview of LLM4Rerank and its principal components.

### 2.1 Problem Formulation

The reranking task plays a pivotal role in recommender systems. As depicted in Figure 1, consider $U$ as the set of users and $I$ as the set of items available for recommendation. For clarity, this paper represents each user and item by a feature vector ($u$ and $i$, respectively). Initially, a ranking model generates a candidate item list $I^r = \{i_1^r, ..., i_n^r, ..., i_N^r\}$ with $N$ items for each user. To improve recommendation performance, a reranking process [21, 24] is applied to analyze the relationships among items within the initial list $I^r$. This analysis aims to generate a refined list with $K$ items, denoted as $I^{re} = \{i_1^{re}, ..., i_k^{re}, ..., i_K^{re}\}(K < N)$, from the initial list. The goal of reranking models is to enhance user-item relevance by optimizing a defined objective function:

$$I^{re} = \underset{i \in I^r}{TopK} R(u, i), \quad (1)$$

where $R(u, i)$ is the scoring function that evaluates the relevance of an item $i$ for a user $u$, considering aspects such as accuracy, diversity, and fairness. Through this optimization, the reranking model selects the optimal recommendation list $I^{re}$ from the candidate lists $I^r$, tailored to individual user preferences, thereby significantly enhancing the quality of recommendations.

In this study, to facilitate equitable comparisons across various reranking baselines, the Generalized Matrix Factorization (GMF)

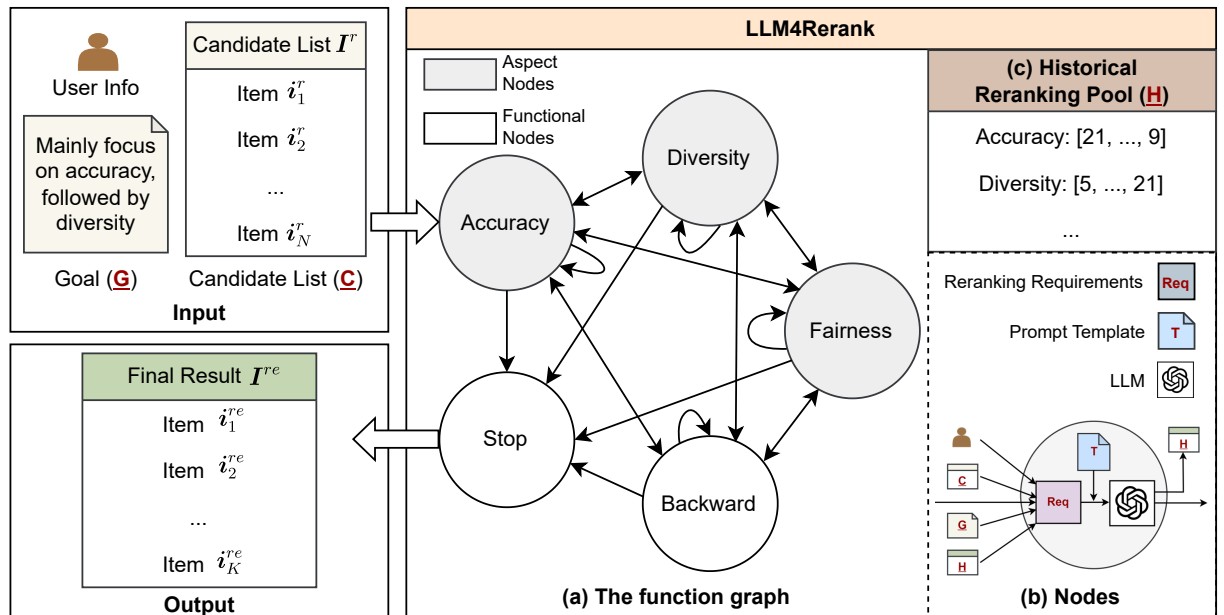

**Figure 2: Overall structure of LLM4Rerank. Inputs are first directed to the "Accuracy" node, which initiates an automatic reranking process in (a). Nodes (b) with varying colors represent distinct aspects or functional steps, guiding the LLM in its deliberations with historical information in (c). A complete reranking process is considered finished once LLM reaches the "Stop" node. For simplicity, items in this figure are represented by their IDs, and the detailed descriptions are hidden.**

model [14] is employed as the uniform global ranking model following previous study [19]. This approach ensures that all reranking models operate on identical candidate item lists.

## 2.2 LLM4Rerank Overview

In this section, we introduce the proposed reranking framework, LLM4Rerank, illustrated in Figure 2. The framework receives three types of inputs: user information (user info), which includes features such as gender and age; the candidate item list $I^r$; and a sentence termed "Goal", which outlines the prioritized aspects for reranking. LLM4Rerank is structured as a fully connected function graph, excluding the "Stop" node, comprising various nodes as shown in Figure 2 (a). Each node represents a potential reranking step that generates a reranking list by LLM, taking into account specific aspect-related or functional requirements as depicted in Figure 2 (b). Each edge within the function graph signifies a potential pathway for node-to-node transition, ensuring connectivity among all nodes, with the exception of the "Stop" node. To exemplify LLM4Rerank's scalability, we integrate not just an "Accuracy" node but also "Diversity" and "Fairness" aspect nodes into the framework, alongside two functional nodes: "Backward" and "Stop" for practical functionalities. The node architecture is meticulously crafted to permit the LLM to sequentially evaluate diverse nodes, thereby optimizing the reranking outcome to fulfill multiple aspect requirements comprehensively. Moreover, to prevent memory loss and enhance the LLM's assessment of aspect combinations, a historical reranking pool is utilized (Figure 2 (c)). This pool records the outcomes from each node in sequence, serving as an auxiliary reference for subsequent reranking at each node. Ultimately, when the "Stop" node is reached, the reranking process is completed. The output at this

stage is precisely the latest reranking results from the historical reranking pool, represented as $I^{re}$.

## 2.3 Nodes Construction

To facilitate the Large Language Model (LLM)'s systematic analysis of complex aspect requirements in reranking, this structure aims to establish distinct nodes for specific requirements. Such an arrangement enables the LLM to process these requirements in a Chain-of-Thought approach [3, 35, 40]. Nevertheless, this approach presents two primary challenges: First, customizing the node structure to maintain scalability when incorporating additional requirements; and second, empowering the LLM to automatically select its subsequent reranking step. To address these challenges, we introduce a generic node structure. It comprises a reranking step, paired with an ancillary indicator that signifies the direction of the forthcoming step identified by the next node's name. This configuration permits the LLM to automatically navigate through the LLM4Rerank framework, making decisions based on the presently available information.

This section presents our strategy to address the challenge of node structure customization and enhance the scalability of the LLM4Rerank framework by introducing a customizable, generic node structure, as depicted in Figure 2 (b). Each node performs a reranking step under LLM considerations, defined by the equation:

$$CN, CR = Function(CN)(\boldsymbol{u}, \boldsymbol{I}^r, Goal, Pool). \qquad (2)$$

Inputs for each node include user information $\boldsymbol{u}$, candidate items $\boldsymbol{I}^r$, a personalized *Goal* sentence for illustrating personalized focus for the entire reranking process, and the historical reranking pool *Pool*, if available. Outputs consist of reranked result $CR$ and the next node's name $CN$ which indicates further reranking steps. The

LLM4Rerank framework crafts prompt specific to reranking criteria and interacts with LLM to generate these outputs.

*2.3.1 Aspect Nodes.* To facilitate the LLM in executing reranking tasks tailored to distinct aspect requirements, we employ a prompt-based template approach within the generic node structure:

$$Function(CN)() = LLM(Temp(CN)()), \qquad (3)$$

where $Temp(CN)()$ is the prompt template for different nodes. This method allows for instantiating specific nodes dedicated to evaluating different aspects within the reranking process. Consequently, each node is designed to systematically address one of these key aspects, ensuring that the reranking outcomes reflect a balanced consideration. In this study, to demonstrate the scalability of LLM4Rerank, we implement three aspect nodes dedicated to reranking: "Accuracy", "Diversity", and "Fairness". Note that the example prompts below are simplified. In practice, detailed prompts should be tailored to input features, aspect characteristics, and evaluation metric semantics.

- **Accuracy Node**: This node is designed to fulfill the performance criteria of the final recommendation list during the reranking phase. As such, the prompt templates are crafted to underscore the correlation between users and items. Figure 3 presents a straightforward template instance employed within the node. Furthermore, given the paramount importance of recommendation accuracy - a fundamental aspect indispensable in recommender systems - the accuracy node has been established as the initial point within the LLM4Rerank framework. Consequently, every reranking procedure commences with the accuracy node, ensuring a foundational focus on precision from the outset.

> An Example of Accuracy Node Template
>
> Considering a user: {User info}
> Here's a list of the candidate movies: {Candidate List}
> Your reranking goal: {Goal}
> Your historical reranking: {Historical Reranking Pool}
> Now, you need to focus on the accuracy aspect (the match between the user and items) and rerank the candidates based on the given information, and then give suggestions about the next step of reranking from the following reranking nodes considering the goal: {Available Nodes}
> For your response format: {Format Description}

**Figure 3: Example prompt template of the accuracy node.**

> An Example of Diversity Node Template
>
> Considering a user: {User info}
> Here's a list of the candidate movies: {Candidate List}
> Your reranking goal: {Goal}
> Your historical reranking: {Historical Reranking Pool}
> Now, you need to focus on the diversity aspect (more items with different xx features should exist at the top of the reranking list) and rerank the candidates based on the given information, and then give suggestions about the next step of reranking from the following reranking nodes considering the goal: {Available Nodes}
> For your response format: {Format Description}

**Figure 4: Example prompt template of the diversity node.**

- **Diversity Node**: This node is specifically designed to address the diversity criteria for the final recommendation list during the reranking phase. In this study, we assess the diversity of the reranking outcomes by evaluating the extent to which a particular attribute of the items is varied within the final list. We employ the $\alpha$-NDCG metric [7] for this purpose. Consequently, an illustrative example of the template used in the diversity node is depicted in Figure 4.

- **Fairness Node**: This node is designated to meet the fairness objectives within the final recommendation list at the reranking phase. In our study, fairness of the recommendation outcomes is operationalized as the average score disparity across two sample groups, segregated by a distinct characteristic, and evaluated using the Mean Absolute Deviation (MAD) metric [44]. Given that the LLM inherently generates reranking lists rather than numerical scores, we allocate scores ranging linearly from 1 to 0 to the items in the final recommendation list. These scores are subsequently utilized to compute the MAD for fairness assessment. For an in-depth methodological exposition, readers are directed to Section 3.1.3. Figure 5 provides a straightforward template illustration for the fairness node.

> An Example of Fairness Node Template
>
> Considering a user: {User info}
> Here's a list of the candidate movies: {Candidate List}
> Your reranking goal: {Goal}
> Your historical reranking: {Historical Reranking Pool}
> Now, you need to focus on the fairness aspect (For items with xxx feature value and items with xxx feature value, You should keep the average ranking of the two categories in the candidates similar) and rerank the candidates based on the given information, and then give suggestions about the next step of reranking from the following reranking nodes considering the goal: {Available Nodes}
> For your response format: {Format Description}

**Figure 5: Example prompt template of the fairness node.**

*2.3.2 Functional Nodes.* Recent research has demonstrated the efficacy of reflection in optimizing the output of LLMs [15, 27]. To augment the logical capabilities of LLM4Rerank in the reranking process and introduce specialized functionalities, we have developed two functional nodes specifically aimed at facilitating reflection and termination within the reranking sequence.

> An Example of Backward Node Template
>
> Considering a user: {User info}
> Here's a list of the candidate movies: {Candidate List}
> Your reranking goal: {Goal}
> Your historical reranking: {Historical Reranking Pool}
> Now, you need to give suggestions about the next step of reranking from the following reranking nodes considering the goal: {Available Nodes}
> For your response format: {Format Description}

**Figure 6: Example prompt template of the backward node.**

- **Backward Node**: This node empowers the LLM to selectively ignore a reranking outcome deemed suboptimal during the evaluation of previous reranking efforts. Within this framework,

LLM4Rerank deletes the latest reranking result from the historical reranking pool and advances to the subsequent node as dictated by the LLM's output directives. An illustrative template example of this node's operation is provided in Figure 6.

- **Stop Node**: This node governs the termination of the LLM4Rerank output sequence. When the LLM4Rerank designates this node as the incoming step, it signifies the conclusion of the complete reranking process. Subsequently, this node extracts the most recent reranking outcome from the historical reranking pool, presenting it as the definitive reranking result. Note that since this node only functionally signals the end of reranking and does not require access to LLM, the prompt template is not required for this node.

## 2.4 Automatic Reranking Process

To leverage the LLM for reranking based on a diverse set of aspect requirements, we have designed distinct nodes, each addressing specific aspect criteria. Nonetheless, delineating a predefined path from one node to another for every reranking task is both inefficient and challenging to achieve. Thus, to accommodate unique user preferences and significantly improve personalization, an automatic reranking process has been developed, which mainly consists of the following three sub-processes:

- **Setting "Goal"**: To accommodate personalized requirements and facilitate LLM4Rerank's scalability across varied contexts, a manually entered sentence, referred to as the "Goal," is incorporated as one of the preliminary inputs for each reranking process. As illustrated in Figure 2, the "Goal" indicates the main focus of a specific reranking process. By interpreting the semantic connections between the "Goal" and the respective nodes, LLM is enabled to automatically select the most appropriate nodes for any given reranking task.
- **Automatic Transition Across Nodes**: For each node, upon receiving replies from LLM, LLM4Rerank would obtain the current reranking results, along with an indicator (i.e., the next node name) for the subsequent node, thereby ensuring a fluid and automatic transition across nodes as illustrated in Equation (2).
- **Conditions to Stop Reranking**: To mitigate the risk of prolonged inactivity and to address errors stemming from possible unrecognized semantic inaccuracies in the LLM's responses, two termination criteria have been established within the framework. The first criterion is triggered when the LLM autonomously identifies the "Stop" node as the subsequent step. The second criterion activates upon the LLM's navigation through a predetermined number of nodes, set by a hyper-parameter, excluding the "Backward" node from this count. Fulfillment of either condition marks the completion of the reranking process. Subsequently, this node retrieves and presents the most recent reranking outcome from the historical reranking pool as the definitive result.

By applying these sub-processes, the whole automatic reranking process of LLM4Rerank is established. The overall Algorithm is also provided in Appendix A.

## 3 Experiments

In this section, we conduct experiments on three widely recognized industrial datasets to explore the following research questions:

**Table 1: Statistics of the used datasets**

| Dataset | Interactions | Users | Items |
|---|---|---|---|
| ML-1M | 1,000,209 | 6,040 | 3,883 |
| KuaiRand | 102,433 | 10,494 | 7,583 |
| Douban-Movie | 759,652 | 2,606 | 34,893 |

- **RQ1**: How does LLM4Rerank compare to established reranking baselines across accuracy, diversity, and fairness aspects?
- **RQ2**: Can LLM4Rerank automatically identify and prioritize a specific blend of aspect requirements for reranking tailored to individual preferences?
- **RQ3**: Does LLM4Rerank's automatic reranking framework offer clear benefits over a predetermined reranking pathway?

### 3.1 Experimental Setup

*3.1.1 Dataset.* We conduct our experiments using three widely recognized public datasets: ML-1M [2] [13], KuaiRand (KuaiRand-Pure) [3] [11], and Douban-Movie [4] [42, 43]. For each dataset, we employ the leave-one-out method [2, 10, 14], a widely adopted approach in the literature, for dividing the data into training, validation, and testing sets. According to previous studies [19], we select the Generalized Matrix Factorization (GMF) model as the global ranking model to generate candidate lists for each user, with each set comprising 20 items. To ensure equitable comparisons across deep learning and LLM-based models, we omit features lacking explicit semantic information and exclude users with fewer than five interactions. For deep learning-based models, we utilize a standard embedding technique [12, 31] to transform various features into vector inputs. Conversely, for LLM-based models, the semantic information of the feature (e.g., the feature's name) is utilized as the input. Table 1 presents the statistics of both datasets subsequent to preprocessing.

*3.1.2 Baseline.* In this section, we assess LLM4Rerank's capability to address diverse aspect requirements by comparing it with the following baseline methodologies:

- **GMF** [14] extends matrix factorization into a non-linear framework, serving as the primary global ranking method in this study. The GMF results represent recommendations prior to the application of any reranking process.
- **DLCM** [1] enhances reranking efficacy by employing a recurrent neural network alongside an attention-based loss function to comprehend local ranking dynamics, aiming primarily at improving accuracy within recommendation outcomes.
- **PRM** [24] leverages a transformer architecture with self-attention mechanisms to refine the entire recommendation list by acknowledging the inter-item influences, thereby concentrating on augmenting accuracy.
- **MMR** [4] aims to balance query relevance with the reduction of redundancy in reranked documents, employing a maximal marginal relevance score to bolster the diversity aspect in recommendation outcomes.
- **FastDPP** [6] expedites Determinantal Point Processes (DPP) for Maximum A Posteriori (MAP) inference, facilitating the efficient

---

[2] https://grouplens.org/datasets/movielens/1m/
[3] https://kuairand.com/
[4] https://www.kaggle.com/datasets/fengzhujoey/douban-datasetratingreviewside-information

**Table 2: Overall performance comparison. Symbols "-A/D/F" represent different focuses "Accuracy/Diversity/Fairness" when setting "Goal" in LLM4Rerank. The default LLM backbone is Llama-2-13B. ↑: higher is better; ↓: lower is better.**

| Model | ML-1M | | | | KuaiRand | | | | Douban-Movie | | | |
|---|---|---|---|---|---|---|---|---|---|---|---|---|
| | HR ↑ | NDCG ↑ | α-NDCG ↑ | MAD ↓ | HR ↑ | NDCG ↑ | α-NDCG ↑ | MAD ↓ | HR ↑ | NDCG ↑ | α-NDCG ↑ | MAD ↓ |
| GMF | 0.4156 | 0.1853 | 0.1005 | 0.0613 | 0.4417 | 0.2314 | 0.1627 | 0.1588 | 0.5723 | 0.3150 | 0.2516 | 0.4006 |
| DLCM | 0.5781 | 0.2354 | 0.1378 | 0.0549 | 0.6893 | 0.3080 | 0.1767 | 0.1026 | 0.6827 | 0.4102 | 0.3581 | 0.2619 |
| PRM | 0.6986 | 0.3246 | 0.1653 | 0.0436 | 0.8083 | 0.3904 | 0.1869 | 0.1032 | 0.6979 | 0.4167 | 0.3477 | 0.2509 |
| MMR | 0.4675 | 0.2588 | 0.2104 | 0.0265 | 0.4928 | 0.2606 | 0.1877 | 0.1569 | 0.6538 | 0.3873 | 0.3744 | 0.2539 |
| FastDPP | 0.4719 | 0.2561 | 0.1942 | 0.0263 | 0.5728 | 0.2913 | 0.1882 | 0.0660 | 0.6635 | 0.4038 | 0.3818 | 0.2820 |
| FairRec | 0.4805 | 0.2007 | 0.1243 | 0.0199 | 0.6083 | 0.2761 | 0.1540 | 0.0318 | 0.6771 | 0.4021 | 0.3119 | 0.1752 |
| RankGPT | 0.5584 | 0.2587 | 0.1799 | 0.0564 | 0.6583 | 0.2910 | 0.1557 | 0.1256 | 0.6635 | 0.3967 | 0.3448 | 0.2472 |
| GoT | 0.5730 | 0.2714 | 0.1942 | 0.0486 | 0.7184 | 0.3198 | 0.1788 | 0.1211 | 0.6827 | 0.4135 | 0.3592 | 0.2195 |
| LLM4Rerank-A | **0.7031**\* | **0.3320**\* | 0.2294 | 0.0434 | **0.8252**\* | **0.4229**\* | 0.2032 | 0.1969 | **0.7041**\* | **0.4301**\* | 0.3806 | 0.2446 |
| LLM4Rerank-D | 0.6875 | 0.3292 | **0.2407**\* | 0.0571 | 0.8058 | 0.4143 | **0.2223**\* | 0.0969 | 0.6701 | 0.4019 | **0.3837**\* | 0.2757 |
| LLM4Rerank-F | 0.5584 | 0.2328 | 0.1411 | **0.0193**\* | 0.7282 | 0.3276 | 0.1825 | **0.0271**\* | 0.6598 | 0.3917 | 0.2970 | **0.1696**\* |
| LLM4Rerank-ADF | 0.6364 | 0.3058 | 0.2051 | 0.0250 | 0.8000 | 0.4117 | 0.2163 | 0.0530 | 0.6877 | 0.4105 | 0.3664 | 0.1975 |

production of diverse recommendation sets. This model focuses on the diversity aspect of the recommendation result.

- **FairRec** [36] introduces a fairness-aware recommendation framework that employs decomposed adversarial learning and orthogonality regularization. It aims to alleviate bias concerning sensitive user attributes, thereby fostering fairness in recommendations without compromising overall performance.
- **RankGPT** [28] investigates the application of LLMs in ranking tasks within information retrieval, employing a novel instructional permutation generation method alongside a sliding window strategy. This model is distinguished by its focus on accuracy. Note that, as a zero-shot LLM baseline, the permutation distillation method in the original paper is not implemented.
- **GoT** [3] proposes a graph-of-thought approach to enhance LLMs' prompting efficacy by structuring generated content as a graph. This facilitates synergistic outcomes, thought distillation, and feedback loop integration, aligning LLM reasoning more closely with human cognitive processes. Unlike LLM4Rerank, GoT adheres to predetermined node-to-node inference paths without historical data consideration. By applying a fixed path, "Accuracy-Diversity-Fairness-Stop." GoT serves as a zero-shot LLM baseline focusing on the combination of accuracy, diversity, and fairness aspects in this paper.

*3.1.3 Implementation Details.* In the evaluation of the **accuracy** aspect, we adopt widely recognized metrics: Hit Ratio (HR) and Normalized Discounted Cumulative Gain (NDCG). For assessing the **diversity** aspect, we apply the commonly used metric $\alpha$-NDCG. To evaluate fairness, we use the Mean Absolute Difference (MAD) [44]. The MAD calculation is formalized as:

$$MAD = \left| \frac{\sum R^{(0)}}{\left| R^{(0)} \right|} - \frac{\sum R^{(1)}}{\left| R^{(1)} \right|} \right|, \tag{4}$$

where $R^{(0)}$ and $R^{(1)}$ represent the predicted ratings for two distinct groups, and $\left| R^{(i)} \right|$ denotes the total number of ratings for group $i$. Within the ML-1M dataset, we utilize the "genre" feature for diversity analysis and the "year" feature for fairness assessment, categorizing films into two groups based on their release year: pre-1996 and post-1996 [16, 44]. For the KuaiRand dataset, "upload_type" serves as the criterion for diversity, while "video_duration"-divided into less than 60,000 ms and greater than 60,000 ms-serves for

fairness evaluation. To fine-tune deep learning-based models for optimal performance, we set the learning rate at 0.001 and engage in a grid search to determine the best hyper-parameters. For zero-shot LLM baselines and LLM4Rerank, Llama-2-13B [29] is selected as the default LLM backbone. The LLM4Rerank's reference code is made available for replication purposes [1]. In addition, we also provide hyper-parameter analysis, inference analysis, and reproduction guideline in the Appendix B, C, D, and E.

## 3.2 Overall Performance (RQ1)

This section presents a comprehensive performance comparison of LLM4Rerank against various baselines, as detailed in Table 2. The comparative analysis reveals that:

- DLCM and PRM achieve acceptable performance in terms of accuracy, as indicated by HR and NDCG metrics. PRM, leveraging a transformer architecture for user-item relevance evaluation, surpasses DLCM in accuracy.
- MMR and FastDPP demonstrate effectiveness in enhancing diversity, as quantified by the $\alpha$-NDCG metric. These models excel in diversifying user reranking lists by emphasizing item similarity and list-wide diversity.
- FairRec exhibits strong performance in promoting fairness, measured using the MAD metric. Through the integration of decomposed adversarial learning and orthogonality regularization techniques, FairRec ensures more equitable recommendations across different user groups.
- RankGPT shows commendable performance, underscoring the capability of zero-shot LLMs in reranking tasks. Conversely, GoT, employing a Chain-of-Thought approach, yields superior outcomes by facilitating a sequential analysis of multiple aspects.
- LLM4Rerank, through personalized "Goal" setting and an automatic reranking process, significantly surpasses baselines, validating its comprehensive efficacy. LLM4Rerank adeptly merges various aspect requirements for reranking, illustrating its versatility. While LLM4Rerank-ADF may not lead in any single aspect, its overall balanced performance across all dimensions confirms the advantage of integrating LLMs with an automatic reranking framework. This approach effectively harmonizes different aspect demands via semantic comprehension, delivering optimized results across accuracy, diversity, and fairness.

**Table 3: Aspect Combination Analysis. "Node Used" indicates the average utilization of Accuracy (Acc), Diversity (Div), and Fairness (Fair) nodes. "Fav Path/Prop" represent the most used path/proportion. "Ave Length" is the average reasoning length. "Max Stop Prop" is the proportion of paths that reach the maximum node count $MC$.**

| Model | ML-1M | | Node Used | | | Path Used | | | |
|---|---|---|---|---|---|---|---|---|---|
| | $\alpha$-NDCG ↑ | MAD ↓ | Acc | Div | Fair | Fav Path | Fav Prop | Ave Length | Max Stop Prop |
| DF | 0.2110 | 0.0255 | 21% | 47% | 32% | A-D-F | 12% | 3.13 | 21% |
| D-F | 0.2337 | 0.0416 | 20% | 59% | 21% | A-D-D-F | 19% | 3.31 | 9% |
| F-D | 0.1456 | 0.0242 | 21% | 27% | 52% | A-F-D-F | 18% | 3.39 | 11% |

**Table 4: Ablation Study of LLM4Rerank**

| Model | HR | NDCG | $\alpha$-NDCG | MAD |
|---|---|---|---|---|
| LLM4Rerank-A | 0.7031 | 0.3320 | 0.2294 | 0.0434 |
| LLM4Rerank-H | 0.6410 | 0.3142 | 0.2275 | 0.0496 |
| LLM4Rerank-AR | 0.6413 | 0.3191 | 0.2200 | 0.0464 |
| LLM4Rerank-N | 0.6533 | 0.3079 | 0.2141 | 0.0515 |

## 3.3 Aspect Combination Analysis (RQ2)

In this section, we delve into experiments designed to assess whether LLM4Rerank can automatically tailor its reranking strategy to incorporate a specific blend of aspect requirements, guided by different user-defined "Goal". Our investigation on the ML-1M dataset encompasses tests with LLM4Rerank under varied "Goals" reflecting distinct prioritizations on the diversity and fairness aspects:

- **DF**: Assign equal importance to diversity and fairness aspects.
- **D-F**: Prioritize diversity, with subsequent emphasis on fairness.
- **F-D**: Prioritize fairness, with subsequent emphasis on diversity.

In this experiment, the maximum node count $MC$ is set to 5. The outcomes, presented in Table 3, indicate that LLM4Rerank proficiently adjusts its reranking paths based on different "Goals", facilitating a dynamic, weighted integration of aspect requirements. This capability significantly bolsters the personalization of the reranking process. It is noteworthy that the "Accuracy" node is consistently involved across all reranking outcomes, underscoring that every reranking sequence commences with the accuracy node. This initial step ensures the foundational accuracy in user-item matching is maintained. Moreover, it is noted that in the LLM's favorite path for different "Goals", the prioritized aspects dominate, indicating that LLM4Rerank framework can drive the LLM to think and capture the importance relation of aspects in "Goals", and influence the reasoning focus of the LLM. Additionally, it can be noted that there are few inference paths that end because the inference node reaches its maximum value. This shows that in the current setting with 3 different aspect nodes, 3-4 thinking steps are enough for the LLM to give the result naturally.

## 3.4 Ablation Study (RQ3)

In this section, we undertake ablation studies on the ML-1M dataset to elucidate the impact of LLM4Rerank's various components on overall performance. The experiments aim to dissect the model's architecture by systematically removing certain features, thereby highlighting their individual contributions. We focus on the "Accuracy" aspect as a case study and align our investigation with a specific "Goal": Pay attention to the accuracy aspects. The following variants of LLM4Rerank are considered for comparison:

- **LLM4Rerank-A**: As detailed in Table 2, including all sub-structures and focusing on the accuracy aspect.

- **w/o historical reranking pool (-H)**: Exclude the historical reranking pool, removing the capability to reference previous reranking outcomes.
- **w/o automatic reranking (-AR)**: Adopting a static reranking path of 'Accuracy-Accuracy-Stop'.
- **w/o other aspect nodes (-N)**: Omit all nodes except for "Accuracy" and "Stop" nodes.

The findings from Table 4, allow us to draw several conclusions:

- The absence of the historical reranking pool (LLM4Rerank-H) leads to a marked decline in performance, underscoring the importance of a holistic view in sequential decision-making. This feature enables LLM4Rerank to recall and evaluate previous choices, enhancing the model's strategic depth.
- The removal of the automatic reranking process (LLM4Rerank-AR) results in a significant performance drop, validating the utility of adaptive pathways in addressing diverse aspect requirements. The automatic reranking mechanism allows LLM4Rerank to dynamically determine subsequent steps based on the entirety of current information, thus optimizing the reranking sequence.
- Eliminating other aspects and functional nodes (LLM4Rerank-N) also precipitates a notable decrease in performance. This highlights the value of a comprehensive review mechanism, as facilitated by the "Backward" node, in mimicking human-like decision-making processes. Meanwhile, in comparison with -AR, the LLM can still decide how many times it can access this node before ending the reranking process. The performance improvement verifies that LLM4Rerank can still benefit from dynamic node visit times, even if LLM4Rerank only has a single aspect node.

These results illuminate the critical roles played by LLM4Rerank's substructures in augmenting reranking performance, particularly in tailoring the process to specific aspect focuses. The study underscores the model's sophisticated architecture, designed to flexibly integrate and balance various reranking criteria.

## 3.5 Case Study

In this section, we present a concrete case study to further illustrate how the LLM4Rerank framework works and whether it can really balance different aspects of reranking as shown in Figure 7. In this figure, we report the two most common paths for LLM4Rerank under the different "Goals": The first one (A-D-F) considers accuracy, diversity, and fairness simultaneously; The second one (A-A-B-D) focuses more on the accuracy aspect, followed by the diversity aspect. The evaluation is based on the average result on the specific path. As shown from the first path, according to the guidance of the "Goal", LLM4Rerank passes through the "Accuracy", "Diversity", and "Fairness" nodes, respectively, and then ends reranking. After the

**Figure 7: Case study of LLM4Rerank on ML-1M dataset. The figure shows the most common paths for LLM4Rerank under the two "Goals". The evaluation is based on the average result on the specific path.**

step of diversity reranking, not only the "$\alpha$-NDCG" metric becomes higher, but also the "HR" and "NDCG" metrics. This may be because in the experiment, LLM can synthesize the current reranking considering not only the current aspect but also the historical reranking results. In addition, the positive relation between "$\alpha$-NDCG" and the "NDCG" metrics may also influence both aspect results when considering them together. From the second path, it can be seen that the addition of functional nodes such as "Backward" helps LLM to think more systematically. When it senses that there is almost no change in the diversity aspect after continuous access to the "Accuracy" node, it considers returning to the previous step and setting the next step as the diversity node.

## 4 Related Work

This section offers a review of the current methods of reranking strategies in recommendations.

### 4.1 Reranking in Recommendations

Reranking emerges as a critical post-processing strategy within recommender systems by introducing supplementary criteria to optimize the initial sequence of candidates. For example, the Deep Listwise Context Model (DLCM) [1] employs a recurrent neural network to sequentially process candidate items, thereby accruing contextual insights. Similarly, the Personalized Re-ranking Model (PRM) [24] utilizes the Transformer architecture [30] for encoding, enabling the modeling of extensive inter-item relationships.

Furthermore, the reranking phase is instrumental in addressing varied aspectual requirements, such as diversity [10, 39] and fairness [26, 38]. These aspects are increasingly recognized for their pivotal role in enriching user experience and ensuring that recommendations are congruent with overarching business objectives. For instance, the Maximal Marginal Relevance (MMR) [4] model formulates a reranking schema emphasizing diversity by balancing document-query similarity against inter-document similarity. Conversely, FastDPP [6] innovates with an efficient greedy Maximum A Posteriori (MAP) inference for Determinantal Point Processes (DPP), facilitating the generation of both relevant and diverse recommendation sets. Additionally, FairRec [36] pioneers a fairness-focused

recommendation framework employing decomposed adversarial learning and orthogonality regularization to mitigate biases and promote equity in the reranking process. Moreover, recent advancements in LLMs indicate that LLM excels in reranking tasks with shorter contexts [18, 22, 28, 37] than in tasks with longer contexts. This discovery makes it possible to use LLM for better and more personalized reranking. For instance, DQ-LoRe [37] uses dual queries for exemplar selection in prompting, RankGPT [28] employs a sliding window for ranking with long context, and the Graph of Thoughts [3] framework improves reranking outcomes with a fixed graph [33, 35] which allows for a complicated step-to-step data processing approach.

However, existing research typically focuses on one aspect, "Accuracy", and occasionally includes one more aspect, such as "Diversity". They overlook the need to integrate multiple aspects of different applications. In contrast, LLM4Rerank employs a function graph for automatic reranking, offering scalability and personalization in combining various aspects to suit different situations. Further discussions about future developments are also provided in Appendix F.

## 5 Conclusion

In this paper, we introduce an LLM-based automatic reranking framework designed to enhance recommender systems through auto-reranking. Central to our approach is the development of a generic node structure, which serves to represent various aspect requirements and functions as distinct nodes within the system. This structure facilitates the construction of a function graph that orchestrates the automatic reranking process, complemented by a historical reranking pool that enables retrospective analysis of reranking decisions. Additionally, a "Goal" sentence is utilized to direct the integration of different nodes, ensuring that the framework can dynamically amalgamate multiple aspect requirements. This design allows LLM4Rerank to deliver superior performance, scalability, and personalization in the reranking process. Experimental validation on three widely recognized industrial datasets underscores the efficacy of the proposed LLM4Rerank framework.

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

**Algorithm 1** The whole automatic reranking process of LLM4Rerank

**Input**: User information $u$, Candidate item list $I^r$, the reranking focus $Goal$, Maximum node count $MC$

**Output**: Final reranking result $I^{re}$

**Note**: $Function(a)(b)$ represents the execution of functions in node $a$ with input $b$.

1: Initialize current node name $CN = Accuracy$; Current reranking result $CR = None$; Node count $NC = 0$; Historical reranking pool $Pool = []$.
2: **while** $CN \neq Stop$ **do**
3:     $CN, CR = Function(CN)(u, I^r, Goal, Pool)$
4:     $Pool.append(CR)$
5:     $NC{+} = 1$
6:     **if** $NC \geq MC$ **then**
7:       $CN = Stop$
8:     **end if**
9: **end while**
10: return $Pool[-1]$

## A    Overall Algorithm of LLM4Rerank

In this section, we present the pseudo-code for the LLM4Rerank algorithm, as detailed in Algorithm 1. The entire reranking process is executed automatically by the LLM4Rerank framework, leveraging the capabilities of LLM. Critical actions, such as summarizing the current step information and determining the next access node (line 3), are assessed and executed autonomously by the LLM. The process continues until the LLM identifies the stop node as the subsequent node, or the total number of node visits reaches the specified threshold (line 7). This framework enables the LLM to engage in iterative, multi-step reasoning in a Chain-of-Thought (CoT) manner, ultimately balancing various aspects to deliver optimal reranking results.

## B    Hyper-parameter Analysis

Recent studies have illuminated the challenges LLMs face in comprehensively processing long contexts laden with dense information [20]. As the number of candidate items in a ranking sequence increases, so does the volume of semantic information, potentially overwhelming LLMs. This could explain the diminished efficacy observed when zero-shot LLMs are directly implemented in recommender systems that catalog millions of items. In light of this, in this section, we explore the impact of the hyper-parameter "candidate item number", initially fixed at 20, on the reranking performance within the ML-1M dataset, as demonstrated in Figure 8 with LLM4Rerank-ADF.

The findings indicate a degradation in LLM4Rerank's performance across various aspects as the "candidate item number" escalates. This outcome not only underscores the current limitations of LLMs in parsing long contexts but also reinforces their aptitude for tasks with fewer items and more concise contextual information, such as reranking, over direct application in extensive recommendation or ranking frameworks. This is attributed to reranking's focus on reordering items based on detailed attributes. Concurrently, there are fewer candidates in the reranking phase following

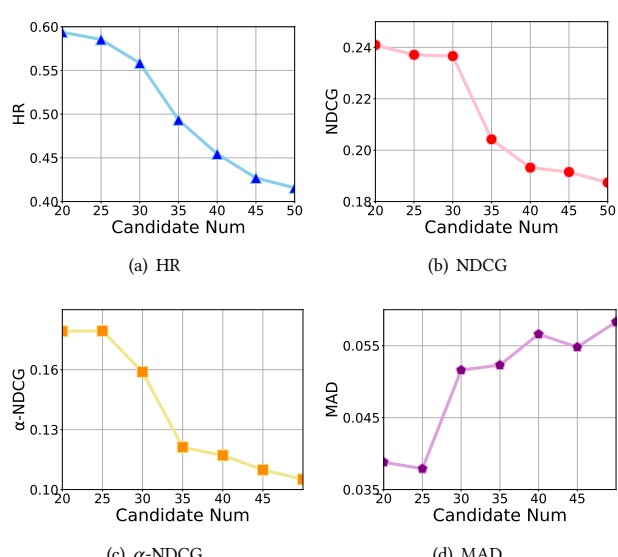

(a) HR      (b) NDCG

(c) $\alpha$-NDCG      (d) MAD

**Figure 8: Hyper-parameter analysis of the "candidate item number" with LLM4Rerank-ADF on ML-1M dataset.**

**Table 5: Inference Analysis**

| Model | RAM | Time per sample |
|---|---|---|
| RankGPT | 14.5GB | 12.74s |
| GoT | 14.5GB | 36.81s |
| LLM4Rerank | 14.5GB | 14.12s |

the initial coarse sorting performed during the ranking phase. By constraining the number of items and focusing on rich semantic content within a single request, LLM4Rerank effectively narrows the semantic chasm across different aspect requirements, thus delivering more coherent reranking outcomes that enhance the overall recommendation quality.

## C    Inference Analysis

Table 5 shows the inference performance of LLM4Rerank compared to other LLM-based models on ML-1M under one of our experiment environments using Llama 2-13B (INT8). Note that since the LLM backbone is replaceable, the inference data provided is only for comparative analysis. As shown in the table, all three models have similar RAM because they use the same LLM backbone and handle only one request at a time. About inference time, LLM4Rerank is slightly more time-consuming than RankGPT because it passes through multiple nodes and requires multiple visits to the LLM, while RankGPT requires only one visit. GoT is the most time-consuming model because it not only accesses multiple nodes but also needs to give multiple sets of answers after each visit to the LLM and aggregate the best results. Moreover, although LLM4Rerank takes a slightly longer inference time than RankGPT, it performs better than RankGPT in all three areas, as mentioned in Table 2 of the main paper. This proves the effectiveness of LLM4Rerank facing multiple aspects.

# D  Guidelines for Reproduction

**Table 6: Used feature fields for ML-1M and KuaiRand**

| Dataset | Feature Fields |
|---|---|
| ML-1M | ['user_id', 'gender', 'age', 'occupation', 'zip', 'item_id', 'title', 'genre', 'year'] |
| KuaiRand | ['user_id', 'user_active_degree', 'is_lowactive_period', 'is_live_streamer', 'is_video_author', 'follow_user_num_range', 'fans_user_num_range', 'friend_user_num_range', 'register_days_range', 'item_id', 'video_type', 'upload_type', 'visible_status', 'server_width', 'server_height', 'music_type', 'author_id', 'music_id', 'video_duration'] |
| Douban-Movie | ['user_id', 'item_id', 'living_place', 'director', 'country', 'language', 'CategoryID'] |

To facilitate the reproducibility of LLM4Rerank, the source code for reference has been made publicly accessible [1]. Our experimental setup was standardized using the NVIDIA GeForce RTX 3060 GPU to ensure consistent computational conditions for our results. It is crucial to emphasize that the hyper-parameters included in the code serve merely as references. The optimal hyper-parameters were identified through an exhaustive grid search process, which is elaborated upon in Section 3.1.3. Furthermore, due to variations in hardware configurations, slight discrepancies in results may occur across different devices, even with identical pseudo-random seed settings, attributed to differences in floating-point computation handling.

In this research, we utilized the ML-1M, KuaiRand and Douban-Movie datasets, upon which a feature selection process was applied to ensure straightforward and equitable comparisons across baseline models. For the ML-1M dataset, in order to facilitate evaluations of diversity and fairness, the "genre" feature was selected to serve as the classification basis for diversity evaluation ($\alpha$-NDCG). Additionally, a "year" feature was extracted to serve as the criterion for fairness evaluation (MAD). In the case of the KuaiRand dataset, to align the inputs for deep learning and LLM-based models as closely as possible, only features with clear semantic significance were selected. The "upload_type" feature was selected to serve as the classification basis for diversity evaluation. The "video_duration" feature was categorized into "short" (less than 60,000 ms) and "long" (more than 60,000 ms) to assess fairness. For Douban-Movie, the "CategoryID" feature was selected to serve as the classification basis for diversity evaluation, and the "language" feature for fairness evaluation. The specific feature fields utilized in both datasets are detailed in Table 6.

# E  Guidelines on adding new nodes

As illustrated in Section 2.3 in the main paper, LLM4Rerank has strong scalability and can easily support the addition of new nodes and do reranking based on different aspects and functions. In this section, we will provide a basic guide to adding nodes.

All of LLM4Rerank's nodes are similar to Figure 2 (b) of the main paper, with multiple structured inputs and two outputs. The inputs contain the user information, the candidate list, the personalized sentence "Goal", and the historical reranking pool (if available). The outputs contain a reranking result, which could be an item ID list

**Figure 9: Example prompt template of the novelty node.**

and an indicator for the subsequent node. In this paper, the indicator is set to be the next node's name. Therefore, to implement new nodes, you should first follow the existing node functions to create a new node function with similar inputs and outputs. Then, a node template should be applied in this function to convert the inputs and the aspect requirements you need into text form. As an example, if you want to consider the "Novelty" aspect in reranking. The simplified template could be as Figure 9. Once you have converted all the information to text using the node template, you can access the LLM through the function and get a reply. After getting the reply, the node function should process the output through the LLM into two parts, one part is the next node's indicator (node name) of the next node, and the other is the current reranking result. At this point, the node function needs to save the current reranking result to the "historical reranking pool" and then proceed to the next node function according to the obtained next node's indicator. In practical applications, in order to better improve LLM output performance. The node prompt can be manually refined according to the aspect characteristics, the realistic significance of the evaluation indicators, and the important relevant features.

# F  Future Directions

In this paper, although LLM4Rerank has showcased the potential of leveraging LLMs for considering multiple aspects in comprehensive reranking, it is currently not superior to traditional models regarding reasoning speed. This limitation stems from the efficiency of the LLM itself rather than the reranking framework as illustrated in Table 5. We remain optimistic that advancements in LLM-related technologies, such as model compression, distillation algorithms, and hardware enhancements, will soon address this issue. Consequently, our future endeavors will focus on enhancing the efficiency of LLM4Rerank while maintaining performance through techniques like compression and distillation. Moreover, as highlighted in Appendix B, current LLMs face challenges in parsing lengthy contexts. To address this, we plan to augment the framework's capability to comprehend extensive text, employing strategies like enhanced Chain-of-Thought (CoT) methods [33] to broaden its applicability.

