# OpenReview forum: "LLM4Rerank: LLM-based Auto-Reranking Framework for Recommendations"
_ACM.org/TheWebConf/2025/Conference — WWW 2025 Oral_

### Official Review · Reviewer_oX92 · 2024-11-25

**Novelty:** 6
**Technical Quality:** 4

**Review:**

### **Summary**:
This paper proposes a LLM-based reranking framework for recommender systems. To balance the complex demands from multiple aspects such as accuracy, diversity and fairness, the authors propose to construct a CoT process which undergoes each of the goals automatically. Extensive experiments on multiple benchmark datasets have shown the effectiveness on various aspects of the proposed framework.

- **Quality**: Overall, this is a well-written paper with detailed content and clear logic. Through extensive experiments and analysis, it is shown that the LLM4Rerank framework has superior performance in many aspects. The author provides theoretical analysis and empirical results, demonstrating the innovation and applicability of the model.
- **Clarity**: The paper is concise in language, reasonable in structure, and clear and intuitive in the methodological part, which helps readers understand the complex framework. However, the introduction of several new concepts (such as function nodes and history pools) is brief, which may cause obstacles for readers who are not familiar with the field.
- **Originality**: The paper proposes a new framework for automated reranking using large language models (LLMs), which is an innovative direction. Through node abstraction and chain of thoughts, the reordering requirements are unified into a semantic space, filling the research gap that existing models struggle to balance in multiple aspects.
- **Significance**:
  - This framework provides a novel approach to integrate multiple requirements (accuracy, diversity, fairness, etc.).
  - This demonstrates the potential of large language models in recommendation systems, particularly in terms of personalization and scalability.

### **Pros**:

- By introducing the LLMs, the problem of traditional reordering models being difficult to balance multiple aspects has been solved.
- A flexible multi node reranking mechanism is achieved through graph structure and "Goal" input.
- The experimental results have demonstrated the superiority of this method on multiple datasets.

### **Cons:**

- Due to the reliance on LLM's reasoning ability, the computational cost is high and may face performance bottlenecks in large-scale applications.
- The paper's measurement and implementation methods for fairness are relatively simple and do not involve more complex practical scenarios.

**Questions:**

- Will there be a performance fall back due to improper node selection? How to avoid such a problem?
- How can historical pools avoid accumulating too much invalid information or noise? How to balance efficiency and effectiveness of historical information for longer sequence node jumps?
- What is the time complexity and computational cost of using LLM for inference at each node? Will such an overhead limit its application in large-scale recommendation systems?
- Does the dataset contain real-world sparsity issues (such as cold start users or long tail recommendations), and how does the model perform in these situations?
- If the target description entered by the user is of low quality or vague (such as "provide good recommendations"), will the performance of the model significantly decrease?

**Reviewer Confidence:**

3: The reviewer is confident but not certain that the evaluation is correct

**Scope:**

4: The work is relevant to the Web and to the track, and is of broad interest to the community

---

### Official Review · Reviewer_Ba34 · 2024-11-27

**Novelty:** 5
**Technical Quality:** 4

**Review:**

Addressing the challenges of existing reranking models in balancing user needs and scalability for personalized demands, this paper introduces a comprehensive large language model-based reranking framework. It integrates multi-faceted user requirements through a universal node structure, chain-of-thought, and prompt templates for true personalization.

Strengths:
1.	The authors present an interesting and concise new framework for personalized recommendations.
2.	The framework automatically fulfills users' personalized needs.
3.	It outperforms several benchmark algorithms.

Weaknesses:
1.	Dependency on large models is unspecified.
2.	Some experimental analyses lack convincing power.
3.	Clarity in descriptions is insufficient.

**Questions:**

1.	Is functional graph construction necessary? Identifying user needs (aspect nodes) seems sufficient.
2.	What is the necessity of deleting the most recent reranking result in Backward node?
3.	The explanation of fairness lacks clarity.
4.	The analysis on adaptive paths (pages 778-782) is not persuasive; comparing only one static path is not statistically significant.

**Reviewer Confidence:**

3: The reviewer is confident but not certain that the evaluation is correct

**Scope:**

4: The work is relevant to the Web and to the track, and is of broad interest to the community

---

### Official Review · Reviewer_Lnnx · 2024-12-01

**Novelty:** 5
**Technical Quality:** 4

**Review:**

This paper studies an important research question: reranking for recommendation. Specifically, the authors propose a framework to leverage large language models (LLM) to rerank while optimizing multiple objectives, including performance/accuracy, diversity as well as fairness. The proposed framework is able to achieve strong performance compared to baselines.

Strengths
- Proposed framework is flexible in terms of ranking objectives and is able to achieve strong performance.
- Comprehensive evaluation results and thorough ablation studies.

Weakness
- A critical part missing from literature as well as this work is the efficiency argument, which can be the bottleneck for LLM as ranking model's success in being deployed to real world applications. The authors should provide a comparison from the efficiency/latency perspective.

**Questions:**

Can you provide an analysis from the efficiency/latency perspective? For example, number of parameters and inference latency compared to baselines.

**Reviewer Confidence:**

3: The reviewer is confident but not certain that the evaluation is correct

**Scope:**

4: The work is relevant to the Web and to the track, and is of broad interest to the community

---

### Official Review · Reviewer_jqhk · 2024-12-02

**Novelty:** 5
**Technical Quality:** 4

**Review:**

The paper proposes a reranking framework for recommender systems leveraging large language models (LLMs). It incorporates multiple objectives simultaneously in the reranking process.

 - **Pros**
   - The methodology itself is quite interesting, and using LLMs and chain-of-thought (COT) reasoning in this context is promising.
   - The experimental results are comprehensive, as the benchmarks include various types of approaches.

- **Cons**
   - The study does not consider the relationships between the multiple objectives. How can the framework balance these objectives effectively?
  - The function graph is treated as a fully connected graph. However, the relationships between different nodes are not explored, and its structure appears overly intuitive. This design choice increases complexity as the number of nodes grows.
   - The paper overlooks existing reranking literature on multi-objective recommendation. Additionally, it does not compare its approach with studies from this domain. For example,
     - Naghiaei, Mohammadmehdi, Hossein A. Rahmani, and Yashar Deldjoo. "Cpfair: Personalized consumer and producer fairness re-ranking for recommender systems." In Proceedings of the 45th International ACM SIGIR Conference on Research and Development in Information Retrieval, pp. 770-779. 2022. (**two fairness objectives and one relevance objective**)
     - Chen, Sirui, Yuan Wang, Zijing Wen, Zhiyu Li, Changshuo Zhang, Xiao Zhang, Quan Lin, Cheng Zhu, and Jun Xu. "Controllable Multi-Objective Re-ranking with Policy Hypernetworks." In Proceedings of the 29th ACM SIGKDD Conference on Knowledge Discovery and Data Mining, pp. 3855-3864. 2023.
  - None of the proposed variants or existing approaches achieve the best performance across all metrics (objectives). This raises questions about the effectiveness of the approach.

**Questions:**

- What is meant by "determine the next node to consider" in the Introduction? This should be explicitly described for clarity.
- The prompt examples provided in the paper and code are quite simplistic, yet prompts play a critical role in the framework's effectiveness. How can prompts be designed to account for multiple objectives effectively?

**Reviewer Confidence:**

3: The reviewer is confident but not certain that the evaluation is correct

**Scope:**

4: The work is relevant to the Web and to the track, and is of broad interest to the community

---

### Official Review · Reviewer_kif5 · 2024-12-04

**Novelty:** 3
**Technical Quality:** 4

**Review:**

This paper proposes to use LLMs for reranking the recommendation results. Experimental evaluations show the efficacy of the proposed LLM4Rerank framework.

Pros:
- the code (not runnable) is available
- the paper is generally easy to follow

Cons:
- The key insight of the proposed method is unclear. Why LLMs could, or are suitable to, solve the mentioned issues in current reranking methods?
- Some improvements in Table 2 are not visually signifcant.

**Questions:**

Why LLMs could, or are suitable to, solve the mentioned issues in current reranking methods?

**Reviewer Confidence:**

3: The reviewer is confident but not certain that the evaluation is correct

**Scope:**

3: The work is somewhat relevant to the Web and to the track, and is of narrow interest to a sub-community